# Controllable cyanation of carbon-hydrogen bonds by zeolite crystals over manganese oxide catalyst

Liang Wang[1], Guoxiong Wang[1], Jian Zhang[1], Chaoqun Bian[1], Xiangju Meng[1] & Feng-Shou Xiao[1]

The synthesis of organic nitriles without using toxic cyanides is in great demand but challenging to make. Here we report an environmentally benign and cost-efficient synthesis of nitriles from the direct oxidative cyanation of primary carbon-hydrogen bonds with easily available molecular oxygen and urea. The key to this success is to design and synthesize manganese oxide catalysts fixed inside zeolite crystals, forming a manganese oxide catalyst with zeolite sheath ($MnO_x@S$-1), which exhibits high selectivity for producing nitriles by efficiently facilitating the oxidative cyanation reaction and hindering the side hydration reaction. The work delineates a sustainable strategy for synthesizing nitriles while avoiding conventional toxic cyanide, which might open a new avenue for selective transformation of carbon-hydrogen bonds.

[1] Key Lab of Applied Chemistry of Zhejiang Province, Department of Chemistry, Zhejiang University, Hangzhou 310028, China. Correspondence and requests for materials should be addressed to L.W. (email: liangwang@zju.edu.cn) or to F.-S.X. (email: fsxiao@zju.edu.cn).

Organic nitriles have been commercially used as common building blocks for high-performance rubbers, polymers and molecular electronics, and are integral parts for producing pharmaceuticals, agrochemicals and fine chemicals, such as vitamins, heterocycles and various carboxylic acid derivatives[1–4]. In 2010, more than 20 additional nitrile-containing pharmaceuticals have been developed in clinical application, demonstrating the greatly important role of organic nitriles[5]. Generally, organic nitriles were synthesized by cyanation of aldehydes using hydrogen cyanide or metal cyanides (for example, KCN, NaCN, $Zn(CN)_2$ and CuCN)[6,7], which are hypertoxic and caused a few environmental disasters around the world (for example, the cyanide spill in Bhopal of India at 1984, and in Baia Mare of Romania at 2000). Therefore, developing green route to avoid the toxic cyanides for production of nitriles is important.

Metal-catalysed direct ammoxidation has been regarded as a sustainable strategy for producing nitriles due to the significant advantage in avoidance of toxic cyanides[1,2,6,8–13]. In these cases, much success has been achieved by employing primary alcohols, aldehydes and aldoximes as precursors for the formation of C-N bonds[1,10–12]. In contrast, the ammoxidation of more easily available hydrocarbon is challenging, because of the high stability of $sp^3$-hybridized C-H bonds[14–20]. In the past few years, homogeneous copper and palladium catalysts have been regarded to be active for C-N formation from C-H bonds to form amides and carbazoles as major products[21–24], although these homogeneous catalysts are difficult to separate and regenerate from the reaction system. The gas-phase ammoxidation of C-H bond over V-Cr oxides is an industrial route, which could easily solve the problems in catalyst separation and also give high selectivity to produce nitriles, but high reaction temperature ($>350\,°C$) and strong corrosivity of gaseous ammonia are still serious problems. Recently, heterogeneous manganese oxide was successfully employed in the ammoxidation of C-H bonds in liquid phase[14], where amides appeared as major products with extremely low selectivity to nitriles. In regard to nitrile synthesis from hydrocarbon feedstocks, the challenge of potentially practical and sustainable routes rely on activating C-H bonds at mild temperatures, as well as hindering the side reactions to selectively form nitriles. Today in the chemical industry, metal catalysts dominate the technology for producing chemicals[25–31], and significant advances in green synthesis processes are mostly combined by the development of new catalysts. The metal catalysts for efficiently oxidative cyanation of C-H bonds are in great demand but extremely difficult to achieve.

We have now made such catalysts by employing nano-sized manganese oxide catalyst fixed inside the silicalite-1 zeolite crystal ($MnO_x@S-1$), where the $MnO_x$ serves as active sites and the microporous S-1 zeolite sheath controls the selectivity by changing the competitive diffusion of water and organic molecules. In this case, the hydrophobic microporous channels of S-1 zeolite are more favourable for the diffusion of organic feed and nitrile product than for the diffusion of water, which hinders the side reaction of nitrile hydration, leading to high efficiency for producing nitriles from various aromatic and aliphatic alkanes. Interestingly, the zeolite sheath also brings the additional advantage of zeolite shape-selective catalysis to the manganese oxide catalyst in oxidative cyanation. This work reports a persuasive example of using a heterogeneous catalyst for oxidative cyanation of C-H bonds to synthesize selectively nitriles in liquid phase, and contributes example of changing the catalytic selectivity of conventional metal oxide catalysts by zeolite crystals.

## Results

**Synthesis and characterization.** The model of $MnO_x@S-1$ catalyst and the catalytic strategy are presented in Fig. 1. Our strategy to synthesize $MnO_x@S-1$ catalyst is based on the solvent-free process recently developed for synthesizing zeolites, which facilitates fixing metallic nanocrystals into zeolite crystal for constructing host–guest structure[32]. The $MnO_x@S-1$ catalyst was synthesized from grinding the solid raw material mixture of tetrapropylammonium hydroxide (TPAOH) and hybrid $SiO_2$-$MnO_x$, followed by thermal treatment at $180\,°C$ for 2 days (Fig. 2a). The manganese loading in the final $MnO_x@S-1$ sample was at 2.1 wt% by inductively coupled plasma (ICP) spectrometer analysis. For comparison, the S-1 zeolite supported $MnO_x$ by conventional impregnation method was also synthesized, which is denoted as $MnO_x/S-1$ with Mn loading at 2.4 wt%.

The successful synthesis of zeolite structure is confirmed by the X-ray diffraction (XRD) patterns and $N_2$ sorption isotherms (Supplementary Figs 1 and 2). Figure 2b–e show the tomogram-section transmission electron microscopy (TEM) images of $MnO_x@S-1$ and $MnO_x/S-1$, which offer the sectioned view of the sample to avoid the influence of $MnO_x$ particles on the external surface. In the images of $MnO_x@S-1$, the $MnO_x$ particles with obviously darker contrast than zeolite could be directly observed, confirming that they are indeed fixed inside the zeolite crystals (Fig. 2b,c, see the TEM images of pure S-1 in Supplementary Fig. 3). In contrast, the images of $MnO_x/S-1$ give $MnO_x$ nanoparticles only on the side of the S-1 crystals (Fig. 2d,e), confirming that they are located on the external surface of S-1 zeolite.

Furthermore, the structure of $MnO_x@S-1$ was investigated by infrared spectra of the adsorbed probing molecule of 2,4-dimethylquinoline (Fig. 3a–c). As presented in Fig. 3b,c, the infrared spectrum of 2,4-dimethylquinoline adsorbed on $MnO_x/S-1$ gives characteristic peaks at 460 and $1,503\,cm^{-1}$. The peak at $460\,cm^{-1}$ is assigned to the Mn-N bond, and the peak at $1,503\,cm^{-1}$ is assigned to a redshift from the vibration of conventional C=N bond in 2,4-dimethylquinoline molecule at $1,510\,cm^{-1}$. These results indicate the presence of strong $MnO_x$–2,4-dimethylquinoline interaction. In contrast, these characteristic peaks are absent in the case of 2,4-dimethylquinoline adsorbed on $MnO_x@S-1$, suggesting the lack of $MnO_x$–2,4-dimethylquinoline interaction. This phenomenon is assigned to that 2,4-dimethylquinoline is inaccessible to $MnO_x$ particles due to the successful encapsulation of the $MnO_x$ inside the

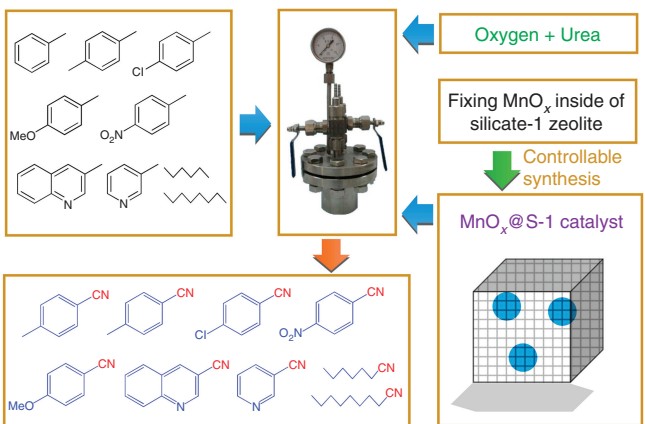

**Figure 1 | Catalytic strategy.** The hydrocarbons with $sp^3$-hybridized C-H bonds and corresponding nitrile products, high-pressure autoclave reactor and the model of $MnO_x@S-1$ catalyst.

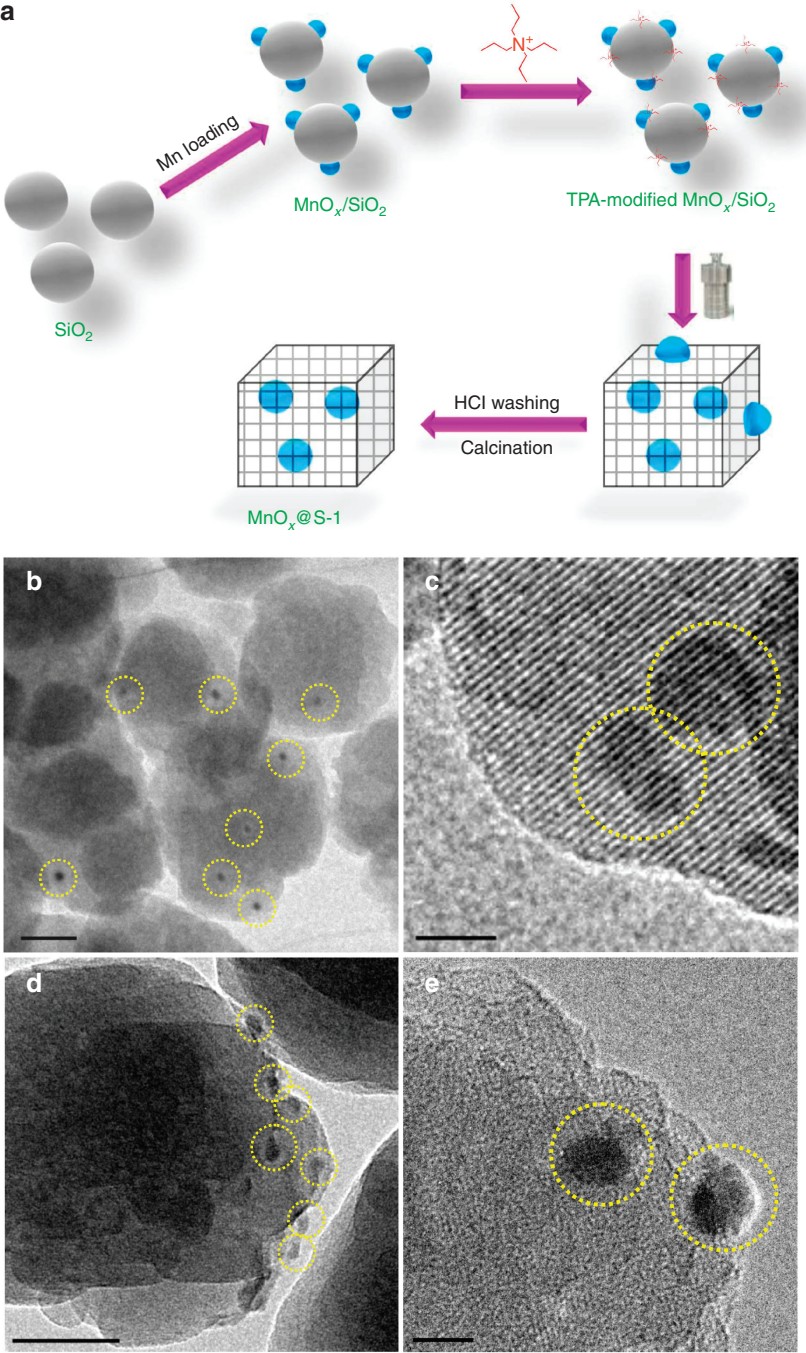

**Figure 2 | Synthesis procedure and TEM characterization.** (**a**) Scheme for the synthesis of MnO$_x$@S-1. Tomogram-section TEM images of (**b,c**) MnO$_x$@S-1 and (**d** and **e**) MnO$_x$/S-1. Scale bar, 50 nm for (**b** and **d**) and 10 nm for (**c** and **e**). The yellow cycles highlight the MnO$_x$ particles.

zeolite crystals. On the contrary, MnO$_x$ on the external of S-1 crystal in the MnO$_x$/S-1 easily accesses to 2,4-dimethylquinoline molecule (Fig. 3a).

Moreover, the structure of the MnO$_x$@S-1 was studied by the aerobic oxidation of probing molecules of benzyl alcohol and 3,5-dimethylbenzylalcohol with different molecular diameters. Figure 3d,e shows the catalytic data over MnO$_x$@S-1 with S-1, MnO$_x$/S-1 and MnO$_x$/SiO$_2$ as reference catalysts. The S-1 zeolite without Mn species is inactive for the reaction. Notably, all the MnO$_x$@S-1, MnO$_x$/S-1 and MnO$_x$/SiO$_2$ are very active for the oxidation of benzyl alcohol with conversion at 86.7–100% (Fig. 3e). For the oxidation of 3,5-dimethylbenzylalcohol, the

MnO$_x$/SiO$_2$ and MnO$_x$/S-1 are still active with conversion at 68.5–78.5%, but the MnO$_x$@S-1 exhibits extremely low conversion (Fig. 3d). This phenomenon is because the 3,5-dimethylbenzylalcohol has larger molecule diameter than S-1 micropore size and is inaccessible to MnOx, causing molecular size selectivity over MnO$_x$@S-1 catalyst.

The combined characterizations of 2,4-dimethylquinoline-adsorbed infrared spectra and TEM characterizaiton, as well as the probing-molecule-oxidation tests, indicate the successful fixing of MnO$_x$ particles into the S-1 crystals. Nonetheless, both MnO$_x$@S-1 and MnO$_x$/S-1 have MnO$_x$ species with similar particle sizes and redox state, as confirmed by TEM images,

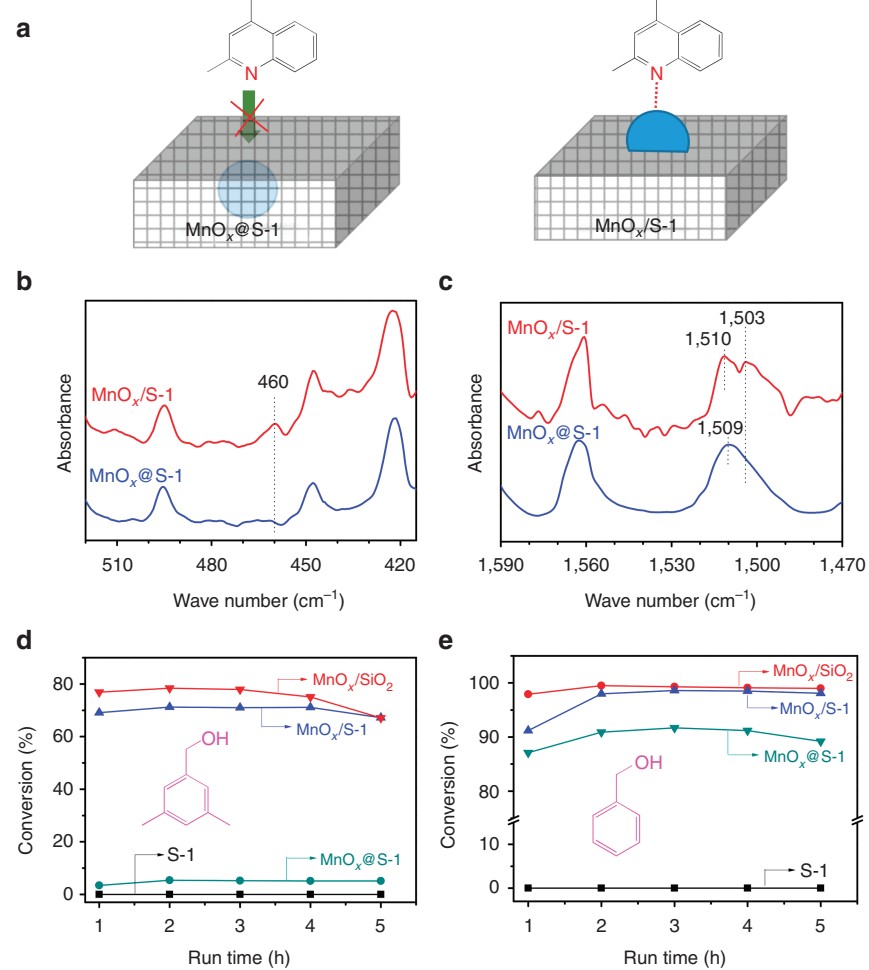

**Figure 3 | Understanding the catalyst structure by probe molecules.** (**a**) Models of 2,4-dimethylquinoline adsorbed on $MnO_x@S$-1 and $MnO_x/S$-1; (**b,c**) 2,4-dimethylquinoline-adsorbed infrared spectra of different samples; (**d**, **e**) catalytic data in oxidation of 3,5-dimethylbenzylalcohol and benzyl alcohol over various catalysts.

ultraviolet–visible spectra, temperature-programmed reduction by $H_2$ and temperature-programmed desorption of $O_2$ tests (Fig. 2 and Supplementary Figs 4–6).

**Catalyst evaluation**. The study on catalytic properties of $MnO_x@S$-1 started from the oxidative cyanation of toluene to benzonitrile in liquid phase (Supplementary Tables 1 and 2 and Supplementary Fig. 7), where toluene acts as model molecule with $sp^3$-hybridized C-H bonds, and abundantly available gaseous oxygen and urea were employed as oxidant and nitrogen source, respectively. A series of supported and homogeneous Mn samples was also tested in the reactions as reference catalysts. As summarized in Table 1, the S-1 zeolite (without loading Mn) is completely inactive for the reaction (entry 1). $MnO_x/S$-1 catalyst is active, giving turnover frequencies (TOFs) at $41.2\,h^{-1}$ in conversion of toluene to benzonitrile, benzamide, benzyl alcohol, benzaldehyde and benzoic acid detected after reaction (Supplementary Table 2). Notably, the yield of benzonitrile was only at 4.5% (entry 2) based on the amount of nitrogen species in the starting sources, while benzamide appears as a major product with a high yield at 84.1% (Supplementary Fig. 8B). Similar phenomenon was also observed on the $MnO_x/SiO_2$ catalyst (entry 3), which are in agreement with the reported phenomenon[14]. In comparison with $MnO_x/S$-1, the

$MnO_x@S$-1 exhibits slightly lower activity, but very different product selectivity with benzonitrile as a major product (Supplementary Fig. 8A), giving the yield at as high as 88.6% (entry 4). Even when air was used as the oxidant, $MnO_x@S$-1 still exhibits high yield of benzonitrile at 72.7% in a prolonged reaction time (entry 5). The homogeneous $Mn(CH_3COO)_2$ catalyst gives very low yield of both benzonitrile and benzamide, giving benzyl alcohol, benzaldehyde and benzoic acid as major products (entry 6), while $KMnO_4$ fails to catalyse the conversion of toluene (entry 7).

Gaseous ammonia, which is mostly used in the industrial cyanation process, was also used as a nitrogen source instead of urea in the $MnO_x@S$-1-catalysed oxidative cyanation reactions. In these tests with a series of hydrocarbon substrates over the $MnO_x@S$-1 catalyst (Supplementary Table 3), the nitrile yields with gaseous ammonia as a nitrogen source are comparable with those with urea as the nitrogen source under the same reaction conditions (Table 2). Additionally, when ammonium hydroxide (23% aqueous solution) was used in the oxidative cyanation of toluene over $MnO_x@S$-1 (Supplementary Table 4), the benzonitrile yield was 62.0% with benzamide yield at 28%, which might be due to the introduction of extra water promoting the formation of benzamide. The introduction of ammonium carbonate and ammonium bicarbonate leads to the formation of benzonitrile with relatively lower yields at 31.1 and 45.6%,

**Table 1 | Catalytic oxidative cyanation of toluene to benzonitrile.**

| Entry | Catalyst | TOF (h$^{-1}$)* | Yield (%)† | |
|---|---|---|---|---|
| | | | **Benzonitrile** | **Benzamide** |
| 1 | S-1 | <1.0 | — | — |
| 2 | MnO$_x$/S-1 | 41.2 | 4.5 | 84.1 |
| 3 | MnO$_x$/SiO$_2$ | 44.0 | 1.1 | 90.8 |
| **4** | **MnO$_x$@S-1** | **32.5** | **88.6** | **<1.0** |
| 5 | MnO$_x$@S-1, in air‡ | 29.1 | 72.7 | <1.0 |
| 6 | Mn(CH$_3$COO)$_2$ | 12.1 | 4.4 | 1.2 |
| 7 | KMnO$_4$ | <1.0 | <1.0 | <1.0 |
| 8 | MnO$_x$/S-1, 30.6 mg of H$_2$O | 44.9 | 1.5 | 85.5 |
| 9 | MnO$_x$@S-1, 30.6 mg of H$_2$O | 40.0 | 85.0 | 2.1 |
| 10 | MnO$_x$@S-1, 2nd run | 34.0 | 88.0 | <1.0 |
| 11 | MnO$_x$@S-1, 8th run | 28.4 | 86.9 | <1.0 |

Reaction conditions: 35 mg of catalyst, 28 mmol of toluene and 0.5 mmol of urea at 160 °C for 4 h with 1.5 MPa of O$_2$. The carbon balance values are over 98% for all cases. The bold entry highlights the best catalytic performances in the table.
*TOF, turnover frequencies calculated based on the total amount of metal atoms in the reaction system at a reaction time of 0.5 h.
†Determined based on the amount of nitrogen in the feed, n-dodecane was used as internal standard, the by-products are benzaldehyde, benzyl alcohol, benzonic acid and some others.
‡3.5 MPa of air was used instead of pure O$_2$, reaction time at 12 h.

because a large amount of benzoic acid are formed (Supplementary Table 5). These data suggest that MnO$_x$@S-1 is efficient for the oxidative cyanation using urea and ammonia. Considering that urea has significant advantages including low corrosivity, relative safety in storage/transportation and easy operation, it was employed as a nitrogen source for the studies in the oxidative cyanation.

The MnO$_x$@S-1 catalyst is reusable. After each reaction run, the catalyst can be easily recycled by filtration with negligible Mn leading as confirmed by ICP optical emission spectrometer. Additionally, it gives constant catalytic performances after the several recycles. In the oxidative cyanation of toluene, for example, in the eighth run, the MnO$_x$@S-1 gives benzonitrile yield at 86.9% with TOF at 28.4 h$^{-1}$ (entry 11 in Table 1), which are comparable to the as-synthesized catalyst, indicating the good recyclability of MnO$_x$@S-1 catalyst.

Table 2 presents catalytic data in oxidative cyanation of various hydrocarbons over the MnO$_x$@S-1 and MnO$_x$/S-1 catalysts. Interestingly, the MnO$_x$@S-1 is efficient in oxidative cyanation of C-H bonds in various toluene-derived methylarenes including p-xylene, 4-chlorotoluene, 4-methylanisole, p-nitrotoluene, 4-bromotoluene, 1-methyl-4-(trifluoromethyl)benzene, 4-methyl-propiophenone, methyl 4-methylbenzoate, ethyl 4-methylbenzoate and m-xylene with moderate to high yields (40.6–83.1%), as well as primary C-H bonds in methylpyridine, 3-methylquinoline and 2-methylthiophene with nitrile yields at 82.5%, 51.4% and 54.1%, respectively. These data suggest the generalized route for oxidative cyanation of a series of aromatic substrates, as well as 3-methylpyridin, 3-methylquinoline and 2-methylthiophene, with primary C-H bonds over the MnO$_x$@S-1 catalyst. In contrast, the MnO$_x$/S-1 catalyst always displays poor yields of nitriles but gives amides as major products (Supplementary Table 6).

Compared with the aromatic substrates, the C-H bonds in aliphatic substrates have higher bonding energy and stability, making their activation challenging. For the conventional liquid-phase oxidation of aliphatic alkanes with molecular oxygen, the successful catalysts display relative low product yields (<10%)[33–35]. Interestingly, the MnO$_x$@S-1 is even active for the oxidative cyanation of aliphatic alkanes such as n-hexane

and n-octane with primary nitrile yields at 19.2% and 6.8%, respectively (Table 2). In contrast, the MnO$_x$/S-1 catalyst gives poor yields of hexanenitrile and octanenitrile at 1.8% and lower than 1.0%, respectively. Additionally, the ZSM-5 zeolite supported Au nanoparticles (Supplementary Fig. 9A), carbon nitride supported Pd nanoparticles (Supplementary Fig. 9b), which are reported to be highly efficient catalysts for the oxidation of aliphatic alkanes[36,37], as well as CoCl$_2$ and Co(CH$_3$COO)$_2$, which are industrial catalysts for aliphatic alkane oxidation, all display low yields of nitriles in the oxidative cyanation of n-hexane and n-octane (<1.5%; Supplementary Table 7). These data suggest the excellent catalytic performances of the MnO$_x$@S-1. Furthermore, when a small amount of initiator was added in the reaction system, the yields of hexanenitrile and octanenitrile were further improved over MnO$_x$@S-1 catalyst, reaching 30.3% and 11.9% (Supplementary Table 8 and Table 2), respectively.

## Discussion

On the basis of the catalytic results above, the MnO$_x$@S-1 could be regarded as selective catalysts for the oxidation cyanation of C-H bonds in hydrocarbons. Compared with the conventional Mn catalysts (for example, MnO$_x$/S-1), the biggest advantage of MnO$_x$@S-1 is the high selectivity to nitrile products rather than amides. From the general knowledge that nitriles could be transformed into amides by hydration with water over transition metals[38–41], we rationally tested the catalysts with small amount of extra water added in the oxidative cyanation reaction systems. Notably, the benzonitrile yield is extremely low over MnO$_x$/S-1 catalyst in the extra water-containing system (1.5%, entry 8 in Table 1). Interestingly, it is found that the MnO$_x$@S-1 catalyst still gives benzonitrile yield at 85.0% with extra water (entry 9), which might be due to the inhibition of benzonitrile hydration over the MnO$_x$@S-1 (Supplementary Fig. 10). Therefore, we also tested the MnO$_x$@S-1 catalyst in benzonitrile hydration by employing MnO$_x$/S-1 and MnO$_x$/SiO$_2$ as reference catalysts under similar reactions conditions to the oxidative cyanation reaction. As presented in Supplementary Table 9, the MnO$_x$@S-1 always exhibited much lower activity for conversion of

**Table 2 | Catalytic oxidative cyanation of various hydrocarbons over the MnO$_x$@S-1 and MnO$_x$/S-1 catalysts.**

$$R\!-\!CH_3 \xrightarrow[\text{Catalyst}]{O_2,\ CO(NH_2)_2} R\!-\!CN$$

| Catalyst | Yields of various nitriles (%) | | |
|---|---|---|---|
| | (4-methylbenzonitrile) | (4-chlorobenzonitrile) | (4-methoxybenzonitrile) |
| MnO$_x$@S-1 | 71.1 | 59.5 | 83.1 |
| MnO$_x$/S-1 | 7.9 | 4.4 | 4.9 |
| | (4-nitrobenzonitrile, O$_2$N) | (3-cyanopyridine) | (3-cyanoquinoline) |
| MnO$_x$@S-1 | 77.7 | 82.5 | 51.4 |
| MnO$_x$/S-1 | 1.5 | 10.5 | 3.3 |
| | (4-bromobenzonitrile, Br) | (4-trifluoromethylbenzonitrile, F$_3$C) | (4-propanoylbenzonitrile) |
| MnO$_x$@S-1 | 66.6 | 70.5 | 80.0 |
| MnO$_x$/S-1 | 5.8 | 16.4 | 9.5 |
| | (methyl 4-cyanobenzoate) | (ethyl 4-cyanobenzoate) | (2-thiophenecarbonitrile) |
| MnO$_x$@S-1 | 84.5 | 72.2 | 54.1 |
| MnO$_x$/S-1 | 10.1 | 9.3 | 1.9 |
| | (3-methylbenzonitrile) | (CN†, hexanenitrile) | (CN†, octanenitrile) |
| MnO$_x$@S-1 | 40.6 | 19.2 (30.3) | 6.8 (11.9) |
| MnO$_x$/S-1 | 9.3 | 1.8 | <1.0 |

Reaction conditions: 35 mg of catalyst, 28 mmol of substrate, 0.5 mmol of urea, 160 °C, 4 h, 1.5 MPa of O$_2$, the yields calculated from the amount of nitrogen in the feed, and *n*-dodecane as an internal standard.
†185 °C, 80 mg of catalyst, 3.5 MPa of O$_2$, 8 h. The values in the parentheses were yields with adding 15 mg of *tert*-butyl hydroperoxide (TBHP, 50% in *n*-dodecane) as initiator, and diphenyl was used as an internal standard.

benzonitrile to benzamide than MnO$_x$/S-1 and MnO$_x$/SiO$_2$ in commercial toluene solvent or with extra water. Considering these catalysts have similar MnO$_x$ species, it is reasonably suggested that the positive effect in hindering the hydration should be directly attributed to the zeolite-fixed structure rather than other factors.

To study the importance of the zeolite-fixed structure, we performed the TPD of toluene and water over the MnO$_x$@S-1 and MnO$_x$/SiO$_2$ catalysts. In the toluene–TPD curves (Supplementary Fig. 11A), the MnO$_x$@S-1 displayed much higher desorption temperature than the MnO$_x$/SiO$_2$. In the water–TPD curves (Supplementary Fig. 11b), the desorption of water on the MnO$_x$@S-1 is much easier than that on the MnO$_x$/SiO$_2$. These results suggest that toluene is overwhelmingly dominant over water for the competitive adsorption in the zeolite micropores. Possibly, during the reaction, the zeolite micropores are fully filled with toluene, resulting in almost complete inhibition for hydration of benzonitrile.

Compared with conventional multiple routes for producing nitriles, the direct oxidative cyanation of C-H bonds with easily available urea and oxygen over the MnO$_x$@S-1 shows obvious advantages such as simplified procedures and avoidance of toxic metal cyanides (Supplementary Fig. 12). Combining with the high activity, selectivity, good recyclability and wide scope of substrates, the MnO$_x$@S-1-catalysed oxidative cyanation is regarded as an ideal route for producing nitriles from aromatic and aliphatic hydrocarbons.

Meanwhile, it is worth noting that the zeolite sheath with micropores might offer a good opportunity for achieving shape-selective catalysis, which is a crucial and an important advantage for heterogeneous catalysts[42–46]. To approve this hypothesis, we performed the oxidative cyanation of a mixture of toluene and 1,3,5-trimethylbenzene. As presented in Supplementary Table 10, the MnO$_x$@S-1 catalyst displays high efficiency for the conversion of toluene to benzonitrile with yield at 85.0%, but completely inactive for the conversion of

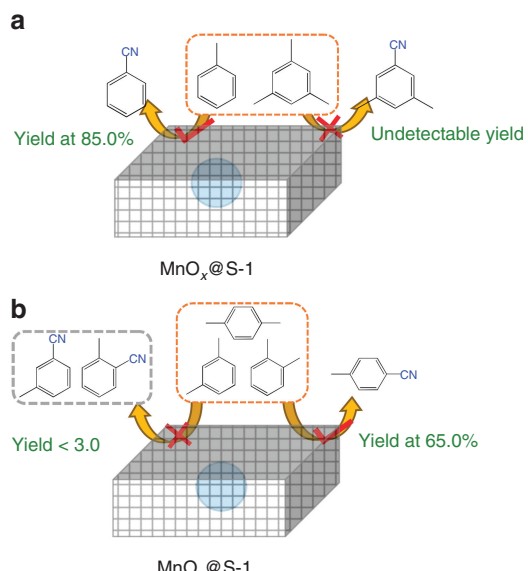

**Figure 4 | Shape-selective catalysis over MnOx@S-1 catalyst. (a)** The MnOx@S-1 is selective for the oxidative cyanation of toluene when using a mixture of toluene and 1,3,5-trimethylbenzene as substrate. **(b)** The MnOx@S-1 is selective for the oxidative cyanation of *p*-xylene when using a mixture of *o*-, *m*- and *p*-xylene as substrate.

1,3,5-trimethylbenzene. This phenomenon is due to that the toluene is small enough to fit the micropores of zeolite S-1, but 1,3,5-trimethylbenzene is too large for accessing the $MnO_x$ sites through the micropores (Fig. 4a). In contrast, the $MnO_x$/S-1 is nonselective for the oxidative cyanation of toluene and 1,3,5-trimethylbenzene, due to the lack of zeolite sheath on the $MnO_x$ sites (Supplementary Fig. 13).

Furthermore, we also performed the oxidative cyanation of *o*-, *m*- and *p*-xylene, which has the same molecular formula but different substituted position of methyl groups. Interestingly, in the catalytic test with a mixture of *o*-, *m*- and *p*-xylene as the substrate (Supplementary Table 11), the $MnO_x$@S-1 displays the yield of *p*-toluonitrile (P1) at 65.0%, but extremely low yields (<3.0%) of *m*-toluonitrile (P2) and *o*-toluonitrile (P3, Fig. 4b). These data are reasonably attributed to the high diffusion coefficient of *p*-xylene in the S-1 zeolite micropores, which is about 100–1,000 times higher than those of *o*- and *m*-xylene[47]. The significantly distinguishable diffusion causes the overwhelming advantage of *p*-xylene in the competitive reaction with *o*- and *m*-xylene, which facilitates the formation of *p*-toluonitrile rather than *o*- and *m*-toluonitrile. All these results demonstrate that the zeolite shape-selective catalysis has been extended to $MnO_x$-catalysed oxidative cyanation by a zeolite sheath in $MnO_x$@S-1 catalyst.

In summary, we have successfully designed and synthesized manganese oxide fixed inside the S-1 zeolite crystals ($MnO_x$@S-1). In the oxidative cyanation of C-H bonds with easily available molecular oxygen and urea, the $MnO_x$@S-1 catalyst exhibited excellent selectivity for the production of nitriles. Additionally, the zeolite sheath also brings the additional advantage of shape-selective catalysis to the $MnO_x$@S-1 catalyst in oxidative cyanation. The approach in this work can be potentially used to develop more efficient heterogeneous catalysts for selective production of nitriles in the future.

## Methods

**Catalyst preparation.** *Synthesis of S-1*: As a typical run, 0.60 g of silica aerogel and 0.51 g of TPAOH (40 wt%) were mixed. After grinding for 20 min, and heating to partly remove the water, the solid powder was transferred into an autoclave to crystallize at 180 °C for 2 days. After calcining at 550 °C for 4 h, the S-1 sample was finally obtained.

*Synthesis of MnOx/S-1 and MnOx/SiO2*: The MnOx/S-1 and MnOx/SiO2 samples were synthesized by wet impregnation using S-1 zeolite and silica aerogel as support, respectively. In a typical run, 3 g of support were added into a 200 ml of aqueous solution containing 1.1 mmol of $Mn(NO_3)_2$ and 27.5 mmol of urea. After stirring at 90 °C for 4 h in a closed reactor to keep away from light, removing the water under vacuum, drying at 100 °C for 12 h and calcining at 480 °C in air for 4 h, the obtained powder was reduced at 400 °C in 10% $H_2$-$N_2$ for 4 h, and calcined in oxygen at 500 °C for 2 h to obtain the final catalyst. By analysis of ICP spectrometer, the Mn loading on MnOx/S-1 and MnOx/SiO2 were calculated at 2.4 wt% and 2.5 wt%, respectively.

*Synthesis of nanosized SiO2-MnOx composite*: As a typical run, 3 g of nanosized SiO2 (mean size at 12–15 nm, available from Xianfeng Nano Co. was dispersed in 800 ml of water under ultrasonic treatment. Then, $Mn(NO_3)_2$ (6.5 mmol), $NH_4NO_3$ (5.5 g) and ammonium hydroxide (25% aqueous solution, 6.2 g) were added and stirred at room temperature for 2 h, and at 90 °C for 4 h in a closed reactor to keep away from light. After removing the water by vacuum, the solid powder were washed with large amount of water, dried at 100 °C for 12 h and reduced at 400 °C in 10% $H_2$-$N_2$ flow for 4 h. Finally, the reduced powder was treated in 5 wt% TPAOH solution (weight ratio of water/ethanol at 1:8) in a reflux system at 90 °C for 12 h, and then the sample was filtrated and dried under vacuum to obtain the TPA-modified SiO2-MnOx composite.

*Synthesis of MnOx@S-1*: MnOx@S-1 was synthesized using TPA-modified SiO2-MnOx composite and silica aerogel as a precursor. As a typical run, 0.35 g of as-synthesized TPA-modified SiO2-MnOx composite and 0.6 g of TPAOH (40 wt%) were mixed. After grinding for 10 min, 0.38 g of silica aerogel was added and continuously grinded for another 20 min. After removing water, the solid powder was transferred into an autoclave to crystallize at 180 °C for 2 days. The obtained powder was washed with hydrochloric acid at 80 °C for 5 min and calcined at 550 °C for 6 h in air to obtain the MnOx@S-1 sample. By ICP analysis, the Mn loading in the sample was 2.1 wt%.

**Characterization.** Powder XRD were obtained with a Rigaku D/MAX 2550 diffractometer with Cu$K\alpha$ radiation ($\lambda = 0.1542$ nm). The metal content was determined using ICP (Perkin-Elmer 3300DV). Nitrogen sorption isotherms were measured at −196 °C using a Micromeritics ASAP 2020M system. The samples were degassed for 10 h at 150 °C before the measurements. TEM images were performed using a Hitachi HT-7700. The samples were ultrasonically dispersed in ethanol and then a drop of the solution was deposited onto a holey C/Cu grid for TEM characterization. To observe clearly the particles encapsulated into the S-1 zeolitic crystals, the image contrast was adjusted. Diffuse reflectance ultraviolet–visible spectra were measured with the PE Lambda 20 spectrometer, and $BaSO_4$ was used as an internal standard sample.

**Catalytic tests.** The oxidative cyanations were performed in a high-pressure autoclave with a magnetic stirrer (1,000 r.p.m.). As a typical run in the oxidative cyanation of toluene, powder catalyst, toluene and urea were mixed in the reactor by stirring for 0.5 h at room temperature. Then, the reaction system was heated to a given temperature (the temperature was measured with a thermometer in an oil bath), oxygen was introduced and kept at the desired pressure (the pressure was measured at the reaction temperature; caution: the experiments involve high pressure of oxygen, which should be performed in avoidance of fire and static electricity). After the reaction, the product was taken out from the reaction system and analysed by gas chromatography (GC-14C, Shimadzu, using a flame ionization detector) with a flexible quartz capillary column coated with free fatty acid phase. The yields of benzonitrile and benzamide were determined based on the amount of urea with *n*-dodecane as an internal standard. The TOFs were calculated from the converted substrate per hour over per molar of Mn species. The recyclability of the catalyst was tested by separation from the reaction system by successive centrifugation, washing with a large quantity of methanol/water and drying at 90 °C for 6 h. Then, the catalyst can be used in the next run. After the sixth run, the catalyst was calcined in oxygen at 400 °C for 4 h.

**Data availability.** Experimental details, N2-sorption, XRD, TEM, ultraviolet-visible, TPD/TPR and more catalytic data are available in the Supplementary Information. All other data are available within the article and its Supplementary Information file or from the authors on request.

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

## Acknowledgements

This work was supported by the National Natural Science Foundation of China (21403192, 91634201, 21333009 and 91645105).

## Author contributions

L.W. performed the catalyst preparation, characterization and performance testing experiments; G.W. and J.Z. performed the catalyst synthesis and performance testing; C.B. and X.M. performed the catalyst characterizations and offered helpful suggestions; L.W. and F.-S.X. planned this study, analysed data and wrote the paper.

## Additional information

**Competing interests:** The authors declare no competing financial interests.

