## [Peer review file · Nature Communications]

Reviewers' comments:

Reviewer #1 (Remarks to the Author):

The synthesis of nitriles from CH₃ groups is a valuable transformation and a challenging one. The work in the manuscript appears to be conducted in a thorough manner, however I do not think the method has the performance which merits it being published in Nature Communications. The method only performs well (in terms of yield) for aromatic substrates (and a limited number are shown in the paper), with aliphatic substrates resulting in low yields. The method requires relatively high temperatures and pressures of O₂ or air. These conditions are all within the explosive regime and if the manuscript is published (somewhere) there should be a note added on safety precautions and equipment used as the conditions are undoubtedly dangerous. The system was optimized for toluene and reactions were carried out neat, and it was found that common organic solvents reduced the performance, so benzene (a very undesirable solvent) is used as the solvent for other substrates. Given all these factors, it does not seem that this method would be viewed as widely applicable by most people.

Reviewer #2 (Remarks to the Author):

The authors report in this study an efficient liquid-phase ammoxidation of alkylarenes using manganese oxide-based catalysts. Nitriles are very important compounds and have been synthesized through various kinds of procedures. For example, benzonitrile and cyanopyridines have industrially been produced by ammoxidation of toluene and methylpyridines, respectively, under rather harsh reaction conditions. Even now, the Sandmeyer reaction and the Rosenmund–von Braun reaction using metal cyanides are two of the most frequently utilized choices for laboratory scale synthesis of aryl and heteroaryl nitriles. In comparison with these procedures, a newly developed procedure in this work is very mild and green procedure. Especially, the selectivity control by the precise catalyst design is very interesting. In my own opinion, this work is interesting, thus I can recommend publishing it in Nature Communication. I think the amide production from alkylarenes is also a very important and interesting transformation. In the present work, this has already been realized when using MnOx/S-1 and MnOx/SiO₂ as the catalysts. In my knowledge, this type of transformation has very rarely been reported. Therefore, if the authors can extend the substrate scope of this amidation reaction, this work may become much more attractive for many readers. I recommend to do this.

Reviewer #3 (Remarks to the Author):

This manuscript reports the synthesis of MnOx inside silicalite-1 crystals for the catalytic ammoxidation of methyl arenes with urea. The MnOx-catalyzed one-pot ammoxidation-hydration to acyl amides was previously reported, and here the zeolitic structure acts as a selective barrier for water in order to avoid the hydration of the intermediate nitrile product. This idea, although not new, is interesting and indeed works well here, with selectivities to nitrile >90%. However, the manuscript is not suitable for Nat. Commun. since key issues need to be addressed before publication could be firmly considered:

- NH₃ should be used as N source, otherwise the process loses interest. Here, only urea and ammonium derivatives are used. Indeed, urea and ammonium carbonate work well while, for instance, ammonium hydroxide does not. May carbonates play a role during the reaction?
- The catalyst restricts the substrate size to monosubstituted methyl arenes, which severely hampers any application in drug synthesis, in contrast to authors' suggestions in the introduction. It could have interest for size selection for small molecules, as exemplified with the results for linear alkanes shown in Table 2. Perhaps with different xylenes?
- Introduction: Zeolite-MnO_x relation is more a host-guest than a core-shell (no complete covering, good accessibility of external reagents). This catalyst is in principle not useful for drug-synthesis. Please tame some sentences, for instance "new perspective for changing..." by "...example of..".
- Discussion: "...regarded as selective catalyst..." (not particularly efficient).
- Table 2, the footnote is not indicated in the Table.
- Many typos, unacceptable for this high-standards journal, please revise. A few: indicating, benzonitrile (many times,...) ming?, MnOAC, fresh, "FFAP)", ...

Point-by-Point Responses to the Comments

Reviewer #1

1-1. Comments: The synthesis of nitriles from CH₃ groups is a valuable transformation and a challenging one. The work in the manuscript appears to be conducted in a thorough manner; however I do not think the method has the performance which merits it being published in Nature Communications. The method only performs well (in terms of yield) for aromatic substrates (and a limited number are shown in the paper), with aliphatic substrates resulting in low yields.

Responses: Thanks for the comments. We agree that the substrate scope is crucial for the catalysis. To provide further evidence for the generality of MnO_x@S-1 catalyst, we studied its catalytic performance in the oxidative cyanation of more aromatic hydrocarbon substrates and 2-methylthiophene. As presented in Reviewer-Only Table 1, the MnO_x@S-1 is effective for the oxidative cyanation of these molecules, displaying medium to high yields of the corresponding nitriles (40.6-84.5%). In contrast, the referenced catalyst of MnO_x/S-1 shows much lower nitrile yields (1.9-16.4%). Combining the data in the original maintext and Reviewer-Only Table 1, it is demonstrated that the MnO_x@S-1 is efficient for the oxidative cyanation with a series of aromatic substrates, as well as 3-methylpyridin, 3-methylquinoline, and 2-methylthiophene (up to 15 substrates).

Compared with the aromatic substrates, the C-H bonds in aliphatic substrates have higher bonding energy and stability. The activation of C-H bonds in aliphatic substrates is always a very challenging topic, which usually requires gas-phase reactions at high temperatures (for example, ~600 °C for propane dehydration) and excessive energy input, resulting in uncontrollable product selectivity and undesirable coke formation (*J. Catal.*, 2005, 233, 234; *J. Catal.*, 2016, 336, 23; *J. Catal.*, 2004, 221, 491; *J. Catal.*, 2000, 192, 128; *Appl. Catal. A*, 2001, 221, 397). For the liquid-phase oxidation of aliphatic alkanes with molecular oxygen, which has

advantages in saving energy at low temperature and reducing the cokes, the successful catalysts displays relative low product yields (<10%, *Angew. Chem. Int. Ed.*, 2000, 39, 2313; *Angew. Chem. Int. Ed.*, 2000, 39, 2310). A well-known example is the industrial oxidation of cyclohexane over Co-/Mn-based catalysts with molecular oxygen, where the cyclohexane conversion were preferentially limited to <5% to minimize the side reactions (*Chem. Rev.*, 2007, 107, 3800; *Nat. Comm.*, 2015, 6, 8466). All these knowledge demonstrates the difficulty in obtaining product yields from aliphatic substrates as high as those from aromatics substrates in the oxidative reactions.

We indeed made great efforts to enhance the yields of nitriles from *n*-hexane and *n*-octane. As presented in Reviewer-Only Table 2, increasing the amount of MnO_x@S-1 (entry 2 in Reviewer-Only Table 2) improved the yield of nitrile products very slightly. However, prolonging the reaction time (entry 3 in Reviewer-Only Table 2) displays a significant enhancement of nitrile yields but being accompanied by decreased carbon balance values. This phenomenon is reasonably assigned to the side reaction of over oxidation—translating hydrocarbons to carbon dioxide (detected by gas chromatography with thermal conductivity detector), which is often observed in the oxidation of hydrocarbons (*Nat. Commun.*, 2015, 6, 8446). Furthermore, we also attempted to enhance the catalysis by the initiator (*e.g.* *tert*-butyl hydroperoxide). It has been well documented that the addition of a small amount of initiator can effectively improve the overall catalytic performances, and this method has been widely used in many oxidation reactions where molecular oxygen is used as a major oxidant (*e.g.* *Nature*, 2005, 437, 1132; *Nat. Comm.*, 2015, 6, 6957; *Chem. Commun.*, 2004, 7, 904; *J. Catal.*, 2011, 281, 30). Interestingly, after addition of a small amount of TBHP as initiator in the reaction system, compare with before, the MnO_x@S-1 gave higher yields of hexanenitrile and octanenitrile at 30.3 and 11.9%, where the carbon balance values were well maintained.

Although the yields of nitriles from aliphatic hydrocarbons were not as high as those from the aromatic substrates, it is considered that the MnO_x@S-1 is a good catalyst for the oxidative cyanation of aliphatic catalyst, because of the well-known difficulty in oxidative transformation of aliphatic hydrocarbons. To provide further

evidence to support this viewpoint, we compared the MnO_x@S-1 with other catalysts. Since there are no heterogeneous catalysts that have been used for the liquid-phase direct oxidative cyanation of C-H bonds in literature, we can not make a direct comparison between our MnO_x@S-1 catalyst and the “state-of-the-art”. To evaluate the catalytic performances of MnO_x@S-1, we prepared some supported catalysts, which are reported in literature to be “highly active” for the oxidation of hydrocarbons (*Chem. Comm.* 2004, 7, 904; *Nat. Commun.* 2015, 6, 8446; *Science* 2011, 331, 195; *Angew. Chem. Int. Ed.* 2015, 54, 13994; *Nat. Comm.* 2013, 4, 1593). These catalysts include carbon supported bimetallic Au-Pd nanoparticles (AuPd/C), Mn-Ce mixed oxides (MnCeO_x), carbon nitride supported Pd nanoparticles (Pd/CN, Reviewer-Only Figure 1B), Al₂O₃ supported Pt nanoparticles (Pt/Al₂O₃), ZSM-5 zeolite supported Au nanoparticles (Au/ZSM-5, Reviewer-Only Figure 1A), Co₃O₄, which are highly active catalysts for the oxidation of aliphatic alkanes in literature, as well as CoCl₂ and Co(CH₃COO)₂, which are industrially catalysts for aliphatic alkane oxidation. These catalysts all display low yields of nitriles in the oxidative cyanation of *n*-hexane and *n*-octane (<1.5%, Reviewer-Only Table 3). In contrast, the MnO_x@S-1 is very active for the oxidative cyanation of *n*-hexane and *n*-octane and exhibits primary nitrile yields at 19.2 and 6.8%, respectively. All these results demonstrate the extraordinary catalytic performances of MnO_x@S-1 in the oxidative cyanation of aliphatic hydrocarbons.

In the revised manuscript, we added the data for more aromatic substrates into Figure 2 of the maintext, and the data for aliphatic hydrocarbons in oxidative cyanation into Table S8 of the Supplementary Information. Considering that the AuPd/C, MnCeO_x, Pt/Al₂O₃ and Co₃O₄ catalysts discussed above have not been thoroughly characterized and they did not exhibit extraordinary performances, we do not include them in the revised manuscript, but show here for review purpose only. The catalytic data of the referenced catalysts, including Pd/CN_x, Au/ZSM-5, CoCl₂ and Co(CH₃COO)₂ are presented in Table S7 of the revised manuscript.

Reviewer-Only Table 1. Catalytic oxidative cyanation of various hydrocarbons over MnO_x@S-1 and MnO_x/S-1 catalysts.

Catalyst	Yields of various nitriles (%)		
			MnO _x @S-1	66.6	70.5	80.0
MnO _x /S-1	5.8	16.4	9.5
			MnO _x @S-1	84.5	72.2	54.1
MnO _x /S-1	10.1	9.3	1.9
			
MnO _x @S-1	40.6		
MnO _x /S-1	9.3		

Reaction conditions: 35 mg of catalyst, 28 mmol of substrate, 0.5 mmol of urea, 160 °C, 4 h, 1.5 MPa of O₂, the yields based on the amount of nitrogen in the feed, and *n*-dodecane as an internal standard.

Reviewer-Only Table 2. Catalytic oxidative cyanation of *n*-hexane and *n*-octane over MnO_x@S-1 under various conditions.

Entry	Reaction conditions	n -hexane		n -octane	
		Yield of hexanenitrile (%)	Carbon balance (%)	Yield of octanenitrile (%)	Carbon balance (%)
1	80 mg of catalyst, 8	19.2	93.2	6.8	97.9

	h				
2	160 mg of catalyst,	19.9	80.1	8.9	93.8
	8 h				
3	80 mg of catalyst,	23.4	65.5	10.2	70.2
	20 h				
4	80 mg of catalyst, 8	30.3	91.5	11.9	97.0
	h, TBHP initiator [†]				

Reaction conditions: 28 mmol of substrate, 0.5 mmol of urea, 185 °C, 3.5 MPa of O₂, the yields based on the amount of nitrogen in the feed, and *n*-dodecane as an internal standard.

[†] 15 mg of *tert*-butyl hydroperoxide (TBHP, 50% in *n*-dodecane) was added as initiator, diphenyl was used as internal standard.

Reviewer-Only Table 3. Catalytic oxidative cyanation of *n*-hexane and *n*-octane over various referenced catalysts.

Entry	Catalyst	n -hexane	n -octane
		Yield of hexanenitrile (%)	Yield of octanenitrile (%)
1	Au/ZSM-5	0.8	0.3
2	AuPd/C	-- [†]	-- [†]
3	MnCeO _x	0.9	-- [†]
4	Pd/CN	-- [†]	-- [†]
5	Pt/Al ₂ O ₃	-- [†]	-- [†]
6	Co ₃ O ₄	0.5	-- [†]
7	CoCl ₂	1.0	-- [†]
8	Co(CH ₃ COO) ₂	1.4	0.9

Reaction conditions: 35 mg of catalyst, 28 mmol of substrate, 0.5 mmol of urea, 185 °C, 4 h, 3.5 MPa of O₂, the yields based on the amount of nitrogen in the feed, and

n-dodecane as an internal standard.

† Undetectable.

Reviewer-Only Figure 1. High-resolution TEM images of (A) Au/ZSM-5 and (B) Pd/CN. The red cycles highlight part of the Au or Pd nanoparticles.

1-2. Comments: *The method requires relatively high temperatures and pressures of O₂ or air. These conditions are all within the explosive regime and if the manuscript is published (somewhere) there should be a note added on safety precautions and equipment used as the conditions are undoubtedly dangerous.*

Responses: We are grateful for the comments. The high-pressure of oxygen or air are widely used in the aerobic oxidation of hydrocarbons (e.g. *Angew. Chem. Int. Ed.*, 2000, 39, 2313; *Angew. Chem. Int. Ed.*, 2000, 39, 2310; *Nat. Comm.* 2015, 6, 6957; *Nat. Comm.* 2015, 6, 8446; *J. Am. Chem. Soc.* 2011, 133, 8074; *Chem. Commun.* 2004, 7, 904). Following the reviewer's suggestion, we have added the caution of "The experiments involve high pressure of oxygen, which should be performed in avoidance of fire and static electricity" in the experiment section of the revised manuscript.

1-3. Comments: The system was optimized for toluene and reactions were carried out neat, and it was found that common organic solvents reduced the

performance, so benzene (a very undesirable solvent) is used as the solvent for other substrates. Given all these factors, it does not seem that this method would be viewed as widely applicable by most people.

Responses: We are very grateful to the referee for pointing this mistake out. We emphasized that no benzene was used in all the experiments, and it should be 28 mmol of substrate in the footnote of Table 2 without any additional solvent. During the revision of the manuscript, all the tables and the corresponding text have been carefully checked.

Reviewer #2

2-1. Comments: The authors report in this study an efficient liquid-phase ammoxidation of alkylarenes using manganese oxide-based catalysts. Nitriles are very important compounds and have been synthesized through various kinds of procedures. For example, benzonitrile and cyanopyridines have industrially been produced by ammoxidation of toluene and methylpyridines, respectively, under rather harsh reaction conditions. Even now, the Sandmeyer reaction and the Rosenmund–von Braun reaction using metal cyanides are two of the most frequently utilized choices for laboratory scale synthesis of aryl and heteroaryl nitriles. In comparison with these procedures, a newly developed procedure in this work is very mild and green procedure. Especially, the selectivity control by the precise catalyst design is very interesting. In my own opinion, this work is interesting, thus I can recommend publishing it in Nature Communication.

I think the amide production from alkylarenes is also a very important and interesting transformation. In the present work, this has already been realized when using $MnO_x/S-1$ and MnO_x/SiO_2 as the catalysts. In my knowledge, this type of transformation has very rarely been reported. Therefore, if the authors can extend the substrate scope of this amidation reaction, this work may become much more

attractive for many readers. I recommend to do this.

Responses: Thanks for the insightful comments. Yes, we agree that amide production from alkylarenes is also important. Following the comments, we tested the catalytic performances of MnO_x/S-1 and MnO_x/SiO₂ catalysts in the amide production from a wide scope of hydrocarbon substrates. As presented in Reviewer-Only Table 4, medium to high yields of amide products were obtained, demonstrating the good catalytic performances of MnO_x/S-1 and MnO_x/SiO₂ in the oxidative amidation. The universality of MnO_x/S-1 and MnO_x/SiO₂ catalysts make them potentially useful for the conversion of various hydrocarbons to the corresponding amides, in good agreement with the reviewer's comments. We have added these important data in Table S6 of the revised manuscript.

Reviewer-Only Table 4. Catalytic oxidative amidation of various hydrocarbons over MnO_x/SiO₂ and MnO_x/S-1 catalysts.

Catalyst	Yields of various amides (%)		
			MnO _x /SiO ₂	60.0	50.5	65.1
MnO _x /S-1	57.9	54.4	65.9
			MnO _x /SiO ₂	78.0	66.6	81.4
MnO _x /S-1	71.5	72.0	73.3
			MnO _x /SiO ₂	56.1	48.3	82.0
			MnO _x /SiO ₂	77.2	78.4	50.1

Reaction conditions: 35 mg of catalyst, 28 mmol of substrate, 0.5 mmol of urea, 160 °C, 4 h, 1.5 MPa of O₂, the yields were determined based on the amount of nitrogen in the feed, *n*-dodecane was used as internal standard.

Reviewer #3

Comments: *This manuscript reports the synthesis of MnO_x inside silicalite-1 crystals for the catalytic ammoxidation of methyl arenes with urea. The MnO_x-catalyzed one-pot ammoxidation-hydration to acyl amides was previously reported, and here the zeolitic structure acts as a selective barrier for water in order to avoid the hydration of the intermediate nitrile product. This idea, although not new,*

is interesting and indeed works well here, with selectivities to nitrile >90%. However, the manuscript is not suitable for Nat. Commun. Since key issues need to be addressed before publication could be firmly considered:

3-1. NH₃ should be used as N source, otherwise the process losses interest. Here, only urea and ammonium derivatives are used. Indeed, urea and ammonium carbonate work well while, for instance, ammonium hydroxide does not. May carbonates play a role during the reaction?

Responses: We are grateful for the valuable comments. Yes, we agree that the nitrogen source is crucial for the reactions, and the ammonia has been widely used in the ammoxidation reactions in gaseous phase at high temperature (>300 °C, *J. Catal.*, 2011, 277, 196; *J. Catal.*, 2006, 243, 350; *J. Catal.*, 2001, 200, 69;). Meanwhile, we note that NH₃ is generally known to be highly corrosive and harmful, which make difficulty in the storage and transportation. Compared with NH₃, urea has significant advantages of relative safety, low corrosivity, and easy operation, which benefits the storage and transportation. In the recent literature on liquid-phase ammoxidation, urea appeared to be used as nitrogen sources instead of gaseous ammonia (*Angew. Chem. Int. Ed.*, 2012, 51, 7250). Therefore, we performed the oxidative cyanation using urea as a nitrogen source in this work.

To address the reviewer's comment, we have performed the oxidative cyanation using gaseous ammonia, and the data over MnO_x@S-1 catalyst are presented in Reviewer-Only Table 5. In these tests with a series of hydrocarbon substrates, the MnO_x@S-1 catalyst exhibits comparable yields to those with urea under the same reaction conditions. These results confirm that the MnO_x@S-1 is active and selective for the oxidative cyanation with either urea or gaseous ammonia as nitrogen sources. These new data using ammonia and the corresponding discussion are provided in the revised manuscript, according to the reviewer's comments.

Additionally, as observed in the new experiments, we found that the carbonate plays a negative role for this reaction. As presented in Reviewer-Only Table 6, the reaction with ammonium carbonate as a nitrogen source gives the benzonitrile yield at

31.1% with a large amount of benzaldehyde/benzoic acid (Reviewer-Only Table 7), which is much lower than that with urea (88.6%). To investigate the role of carbonate species, we artificially added K_2CO_3 into the reactor with urea as the nitrogen source. When a small amount of K_2CO_3 (molar ratio of urea/ K_2CO_3 at 5, entry 3 in Reviewer-Only Table 6) was added in the feed, the $MnO_x@S-1$ catalyst exhibit lower benzonitrile yield at 80.1% than that of the K_2CO_3 -free system (88.6%, entry 1). Further increasing the amount of K_2CO_3 (molar ratio of urea/ K_2CO_3 at 1, entry 4 in Reviewer-Only Table 6) leads to a significant decrease of benzonitrile yield at 47.3% with the formation of a large amount of benzoic acid (Reviewer-Only Table 7). The obvious change in the product selectivity is undetectable with the addition of KCl or CO_2 instead of K_2CO_3 . Based on these results, we suggest that the carbonate species could promote the direct oxidation into benzoic acid, thus hindering the cyanation to benzonitrile.

When ammonium hydroxide (23% aqueous solution) was used as a nitrogen source, the $MnO_x@S-1$ exhibits benzonitrile yield at 62.0%, which is due to the introduction of a large amount of water (~3.2 mmol, exceeding the limitation for hindering the side reaction, as shown in Figure S9 in Supporting Information), leading to the side reaction of hydration to form benzamide (yield at 28.0%).

We have added these data in Tables S3-S5 of the revised manuscript.

Reviewer-Only Table 5. Catalytic oxidative cyanation of various hydrocarbons over MnO_x@S-1 using gaseous ammonia.

Yields of various nitriles (%)		
		78.4	66.4	64.7
		67.3	71.3	84.1
		82.2	56.7	16.6

Reaction conditions: 35 mg of catalyst, 28 mmol of substrate, 1.0 mmol of gaseous ammonia, 160 °C, 4 h, 1.5 MPa of O₂, the yields based on the amount of nitrogen in the feed, and *n*-dodecane as an internal standard.

Reviewer-Only Table 6. Oxidative cyanation of toluene over the MnO_x@S-1 catalyst using various nitrogen sources.

Entry	N sources	TOF (h ⁻¹) [†]	Yield (%) [‡]	
			Benzonitrile	Benzamide
1	urea	32.5	88.6	<1.0
2	(NH ₄) ₂ CO ₃	28.9	31.1	3.3
3	urea + K ₂ CO ₃ [†]	32.3	80.1	<1.0
4	urea + K ₂ CO ₃ ^{&}	35.4	47.3	5.9
5	urea + KCl [#]	29.8	85.9	<1.0

6	urea + CO ₂ ^{\$}	32.2	85.0	<1.0
7	ammonium hydroxide	40.0	62.0	28.0

Reaction conditions: 35 mg of catalyst, 28 mmol of toluene, 0.5 mmol of urea, 160 °C, 4 h, and 1.5 MPa of O₂.

[†]The TOFs were calculated from the total amount of metal atoms in the reaction system at a reaction time of 0.5 h.

[‡]The yields were determined from the amount of nitrogen atom, and *n*-dodecane was used as an internal standard.

[‡]molar ratio of urea/K₂CO₃ at 5.

[&] molar ratio of urea/K₂CO₃ at 1.

[#] molar ratio of urea/KCl at 2.5.

^{\$} 0.5 mmol of CO₂.

Reviewer-Only Table 7. The yield of nitrogen-free by-products in the oxidative cyanation of toluene over the MnO_x@S-1 catalyst using various nitrogen sources.

Entry	N sources	Yield (%) [†]		
		Benzyl alcohol	Benzaldehyde	Benzoic acid
1	urea	4.4	9.0	9.3
2	(NH ₄) ₂ CO ₃	<1.0	9.5	38.9
3	urea + K ₂ CO ₃ [‡]	2.0	16.1	17.2
4	urea + K ₂ CO ₃ ^{&}	<1.0	4.2	33.3
5	urea + CH ₃ COOK [#]	<1.0	13.8	11.0
6	urea + CO ₂ ^{\$}	3.5	8.1	7.3
7	ammonium hydroxide	5.0	10.7	9.9

The reaction conditions are the same to those in Reviewer-Only Table 6.

† For comparison with the yields of benzonitrile, the yields of these nitrogen-free by-products were determined from the amount of nitrogen in the feed, *n*-dodecane was used as an internal standard. Yield = (amount of product) / (amount of nitrogen in the feed)*100%.

[‡] molar ratio of urea/K₂CO₃ at 5.

[&] molar ratio of urea/K₂CO₃ at 1.

[#] molar ratio of urea/KCl at 2.5.

^{\$} 0.5 mmol of CO₂.

3-2. The catalyst restricts the substrate size to monosubstituted methyl arenes, which severely hampers any application in drug synthesis, in contrast to authors' suggestions in the introduction. It could have interest for size selection for small molecules, as exemplified with the results for linear alkanes shown in Table 2. Perhaps with different xylenes?

Responses: We thank the reviewer for raising the insightful comments. Yes, the MnO_x@S-1 is highly efficient for the transformation of small molecules but has limitation for the bulky molecules, this might offer opportunity for achieving shape-selective catalysis. Following the reviewer's suggestion, we performed the oxidative cyanation of toluene and 1,3,5-trimethylbenzene. As presented in Reviewer-Only Table 8, the MnO_x@S-1 catalyst is very active for the conversion of toluene to benzonitrile, but completely inactive for the conversion of 1,3,5-trimethylbenzene, which confirms the shape selectivity of MnO_x@S-1 catalyst. This feature is reasonably attributed to the zeolite sheath, where the toluene is small enough to fit the micropores of zeolite but 1,3,5-trimethylbenzene is too large for accessing to the metal sites (Reviewer-Only Scheme 1A). In contrast, the MnO_x/S-1 shows similar performances in the oxidation of toluene and 1,3,5-trimethylbenzene in individual and mixed tests, whereas the Mn sites in the MnO_x/S-1 without zeolite

sheath are exposed and accessible to both toluene and 1,3,5-trimethylbenzene (Reviewer-Only Scheme 1B).

Reviewer-Only Table 8. Catalytic oxidative cyanation of toluene and 1,3,5-trimethylbenzene over MnO_x@S-1 catalyst.

Catalyst	Product	Yield (%) [†]	Yield (%) [‡]
MnO _x @S-1	P1	85.0	87.8*
	P2	--	--
MnO _x /S-1	P1	4.2	4.5
	P2	47.1	49.5

[†] A mixture of toluene and 1,3,5-trimethylbenzene as the substrate.

[‡] In the individual tests, toluene or 1,3,5-trimethylbenzene was used as the substrate.

* Note that this yield is from an independent catalytic tests from that of entry 4 in Table 1. The results from two independent experiments are well consistent.

Reviewer-Only Scheme 1. Understanding the shape-selective feature of $\text{MnO}_x@S-1$ and $\text{MnO}_x/S-1$. (A) The $\text{MnO}_x@S-1$ is selective for the oxidative cyanation of toluene when using a mixture of toluene and 1,3,5-trimethylbenzene as substrate. (B) The $\text{MnO}_x/S-1$ is active for both molecules.

Reviewer-Only Table 9. Catalytic oxidative cyanation of *o*-, *m*-, *p*-xylene over $\text{MnO}_x@S-1$ catalyst.

Catalyst	Product	Yield (%) [†]
$\text{MnO}_x@S-1$	P1	65.0
	P2	<1.0
	P3	2.9

[†] A mixture of *o*-, *m*-, and *p*-xylene molecules with molar ratio at 1 as the substrate.

Reviewer-Only Scheme 2. Understanding the shape-selectivity of $\text{MnO}_x@S-1$ in oxidative cyanation various xylenes. The $\text{MnO}_x@S-1$ is selective for the oxidative cyanation of *p*-xylene when using a mixture of *o*-, *m*-, and *p*-xylene as the substrate.

Furthermore, we also performed the oxidative cyanation of *o*-, *m*-, *p*-xylene,

which has the same molecular formula but different substituted position of methyl groups. Interestingly, in the catalytic test with mixture of *o*-, *m*-, and *p*-xylene substrates, the MnO_x@S-1 displayed the yield of *p*-toluonitrile (P1) at 65.0%, but extremely low yield (<3.0%) of *m*-toluonitrile (P2) and *o*-toluonitrile (P3), demonstrating the shape-selectivity of MnO_x@S-1 (Reviewer-Only Table 9, Reviewer-Only Scheme 2). These data are reasonably attributed to the high diffusion coefficient of *p*-xylene in the S-1 zeolite micropores, which is about 100-1000 times higher than those of *o*- and *m*-xylene (*J. Catal.*, **1993**, 139, 24-33). The significantly distinguishable diffusion causes the overwhelming advantage of *p*-xylene in the competitive reaction with *o*- and *m*-xylene, which facilitates the formation of *p*-toluonitrile rather than *o*- and *m*-toluonitrile.

In summary, all these results demonstrate the zeolite sheath offers an additional advantage for shape-selective catalysis in oxidative cyanation, and the shape selectivity is regarded as a superior advantage for the heterogeneous catalysts (*Nat. Mater.*, 2017, 16, 132; *Angew. Chem. Int. Ed.*, 2016, 55, 9178; *J. Am. Chem. Soc.*, 2016, 138, 7880; *Nat. Chem.*, 2012, 4, 1030; *J. Am. Chem. Soc.*, 2015, 137, 4276; *Angew. Chem. Int. Ed.*, 2016, 55, 3685). In contrast, the conventional supported catalysts (*e.g.* MnO_x/S-1) are generally non-selective. Following the reviewer's comments, we added these data and discussion on shape selectivity in the revised manuscript, which really improves the quality of this work because of the great importance of shape selectivity in heterogeneous catalysis. Additionally, following the comments, we also modified the sentence about drug synthesis in the section of introduction.

3-3. Introduction: Zeolite-MnO_x relation is more a host-guest than a core-shell (no compete covering, good accessibility of external reagents). This catalyst is in principle not useful for drug-synthesis. Please tame some sentences, for instance "new perspective for changing..." by "...example of..".

Responses: Thanks for the comments. Yes, we agree that the *host-guest* should be

more suitable than *core-shell* to describe our catalysts. We have modified the sentences according to the reviewer's suggestion.

3-4. Discussion: "...regarded as selective catalyst..." (not particularly efficient).

Responses: We thank the reviewer for this suggestion, this sentence has been modified to "*...regarded as selective catalyst...*" in the revised maintext.

3-5. Table 2, the footnote is not indicated in the Table.

Responses: Thanks for the comment. We have shown the footnote in the revised Table 2.

3-6. Many typos, unacceptable for this high-standards journal, please revise. A few: indicating, benzonitrile (many times,...) ming?, MnOAC, fresh, "FFAP)".

Responses: We are very grateful for the reviewer for pointing the mistakes out, which have been carefully corrected in the revised manuscript.

REVIEWERS' COMMENTS:

Reviewer #2 (Remarks to the Author):

In my opinion, the manuscript has greatly improved. The efforts made by the authors to thoroughly respond to all the points addressed by the referees is greatly appreciated. It is now a really nice manuscript, and thus I can fully support its publication.

Reviewer #3 (Remarks to the Author):

Authors have made a great effort and have eventually fulfilled all my requirements. The manuscript is now much more appealing. However, I agree with Reviewer 1 that it is difficult to see a real application of this method either at the lab. or industrial scale. In any case, the results open new lines of attack for this reaction and may be of interest for the readers of Nat. Commun